# On the Data-Efficiency with Contrastive Image Transformation in Reinforcement Learning

**Sicong Liu**$^{1\,2\,3^{*}}$  **Xi Sheryl Zhang**$^{2\,3\,5^{\dagger}}$  **Yushuo Li**$^{2}$  **Yifan Zhang**$^{2\,3\,5}$  **Jian Cheng**$^{2\,3\,4}$

$^{1}$NJUST,  $^{2}$Institute of Automation, Chinese Academy of Sciences,
$^{3}$AIRIA,  $^{4}$School of Future Technology, University of Chinese Academy of Sciences,
$^{5}$University of Chinese Academy of Sciences, Nanjing
`{sicongliu1014;sheryl.zhangxi}@gmail.com`,
`yushuo.li@ia.ac.cn, {yfzhang;jcheng}@nlpr.ia.ac.cn`

## Abstract

Data-efficiency has always been an essential issue in pixel-based reinforcement learning (RL). As the agent not only learns decision-making but also meaningful representations from images. The line of reinforcement learning with data augmentation shows significant improvements in sample-efficiency. However, it is challenging to guarantee the optimality invariant transformation, that is, the augmented data are readily recognized as a completely different state by the agent. In the end, we propose a contrastive invariant transformation (CoIT), a simple yet promising learnable data augmentation combined with standard model-free algorithms to improve sample-efficiency. Concretely, the differentiable CoIT leverages original samples with augmented samples and hastens the state encoder for a contrastive invariant embedding. We evaluate our approach on DeepMind Control Suite and Atari100K. Empirical results verify advances using CoIT, enabling it to outperform the new state-of-the-art on various tasks. Source code is available at `https://github.com/mooricAnna/CoIT`.

## 1 Introduction

Improving data-efficiency to accomplish sequential decisions has always been a crucial problem in pixel-based reinforcement learning. As the agent has to learn an optimal policy with a meaningful information abstraction from observations parallel. Unlike supervised representation learning with strong supervised high-dimensional signals, the training process in RL is fragile. It could be harmful to the training process and cause performance degradation consequently using inappropriate manners. Hence, it is an urgent request to seek subtle representation learning methods for visual RL.

Previous works have been proposed in the literature to demonstrate that introducing auxiliary loss functions such as pixel reconstruction (Yarats et al., 2019) and contrastive learning (Laskin et al., 2020b) alleviates this issue. In particular, data augmentations have already proven beneficial to data-efficiency. RAD (Laskin et al., 2020a) performs an extension of experiments and widely analyzes the impact of various techniques in data augmentation. DrQ (Yarats et al., 2020) and DrQ-v2 (Yarats et al., 2021) make use of appropriate image augmentation with great success. Also, previous works have carried out the potential of data augmentation in terms of generalization (Hansen et al., 2021; Raileanu et al., 2020; Zhang & Guo, 2021; Hansen & Wang, 2021; Fan et al., 2021).

Despite the mentioned efforts, it is pretty hard to guarantee that the augmented representations are sufficiently diverse yet semantically consistent. To this end, we explore the underlying condition for representation learning in RL. It is rational to hypothesize that there is an optimal transformation enabling an encoder to abstract informative latent space. This line of works belongs to the regime of state abstraction (Du et al., 2019; Zhang et al., 2020b; Tomar et al., 2021; Wang et al., 2022), which derives from grouping similar world-states for descriptions of the environment (Dieterich, 2000; Andre & Russell, 2002; Castro & Precup, 2010). Inspired by spatial transformer networks (STN)

---

$^{*}$Work done during Institute of Automation, Chinese Academy of Sciences internship.
$^{\dagger}$Corresponding Author.

(Jaderberg et al., 2015), a data augmentation model in the vision domain, we consider that merging the parameterized transformation with visual RL could be beneficial. The designed transformation not only discovers the optimality of state abstraction but also produces diverse virtual samples for the agent. To do so, we enforce a learnable data augmentation that updates its parameters along with the RL objective.

To understand parameterized augmentation and its relation to representation learning in RL, we focus on fundamental data manipulation by generating augmented data from a learnable Gaussian distribution. To be clear, we present the image transformation to control the margin of the augmentation under an RL training-friendly data distribution. Since changed data distributions meanwhile being controlled by learning algorithms would be helpful in high-dimensional cases (Balestriero et al., 2021). Here we raise our idea:

*Can we parameterize the data augmentation by sampling from a dynamic distribution to obtain a training-friendly state distribution along with model-free RL?*

In light of this challenge, we present a contrastive invariant transformation (CoIT), a novel contrastive learning to ameliorate the data-efficiency for visual RL. CoIT integrates a learnable transformation for model-free methods with minimal modification to the architecture and training pipeline. Specifically, we parameterize the mean and variance of a Gaussian distribution for transforming data and update the parameters together with RL by using constraints to urge faster algorithm convergence empirically. As the learning goes on, the agent approximates the `TRANSFORM` distribution that is optimal for the task at hand to solve the task. In addition, we evaluate CoIT on DeepMind Control Suite and Atari100K, and experimental results demonstrate that the learnable transformation outperforms the current SOTA methods. Besides, our method does not claim any custom architecture choices and is essential for reproducing end-to-end training. Based on these results, we demonstrate that a learnable transformation improves data-efficiency effectively for visual RL.

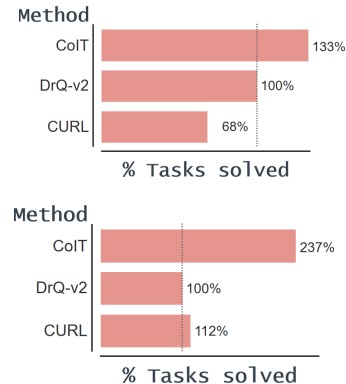

Figure 1: Percentage (%) of score solved in the DMC. We set the score of DrQ-v2 as $100\%$ and report the result of CoIT and CURL: **Up**: in 500K steps. **Bottom**: in 100K steps. The task is solved when return nearly reaches the upper bound.

**Key Contributions:** (i) We present CoIT, a simple yet effective framework with a learnable image transformation that integrates invariant representations with model-free RL to improve data-efficiency. (ii) We propose a theoretical analysis of how our method can approximate a stationary distribution over the transformed data by the optimal invariant metric, thus learning better representations. (iii) We evaluate CoIT on popular benchmarks and show that our method outperforms previous state-of-the-art methods on data-efficiency and stability.

## 2 RELATED WORK

Several concurrent methods have been proposed for improving data-efficiency whose common ingredients containing data augmentation and self-supervised learning are listed.

**Data augmentation in RL.** Like the success of data augmentation in computer vision (Zhong et al., 2020; DeVries & Taylor, 2017; Yun et al., 2019; Zhang et al., 2017), these methods have played a key role in improving the data-efficiency of visual RL problems Mnih et al. (2013); Yarats et al. (2019); Hafner et al. (2019); Lee et al. (2019). RAD (Laskin et al., 2020a) conducted mounts of experiments and finds out that different data augmentations lead to entirely different results. It provides a broader perspective for the follow-up study of data augmentation in RL. DrQ (Yarats et al., 2020) proposed an effective augmentation method called *random shift* and introduced a regularization term for Q-learning. Based on DrQ, the DrQ-v2 (Yarats et al., 2021) conducted minimal changes and demonstrated that merely a simple augmentation method could match the state-of-the-art model-based algorithm on data-efficiency and performance.

**Self-supervised learning in RL.** Motivated by the breakthrough in self-supervised learning (Chen et al., 2020; He et al., 2020; Caron et al., 2020; Grill et al., 2020), it is natural to combine these

methods with visual RL to learn rich representations. CURL (Laskin et al., 2020b) introduces a framework similar to SimCLR (che) into visual RL. CoBERL (Banino et al., 2021) also tried to minimize the consistency between positive samples by semantic-preserving data augmentation. Besides, STDIM (Mazoure et al., 2020) and PI-SAC (Lee et al., 2020) maximize the temporal mutual information (MI) between the nearby states. SPR (Schwarzer et al., 2020) and PlayVirtual (Yu et al., 2021) follow their idea, but they utilize the *dynamics model* to predict nearby states in latent space. DBC (Zhang et al., 2020b) and PSM (Agarwal et al., 2021) focus on learning task-relevant information. They utilize signals in the environment to achieve an invariant representation learning and thereby generalize the agent to unseen environments.

## 3 PRELIMINARIES

### 3.1 REINFORCEMENT LEARNING FROM OBSERVATIONS

Visual RL control is formulated as an infinite-horizon Markov Decision Process (MDP) (Bellman, 1957), as the observations can not fully describe the underlying state. To address this problem, we stack multiple consecutive frames together to represent the current underlying state $\mathbf{s}$ (Mnih et al., 2013). In this mind, the MDP $\mathcal{M}$ is a 5-tuple $\langle \mathcal{O}, \mathcal{S}, \mathcal{A}, r, \gamma \rangle$. Here, the observation space $\mathcal{O}$ generally consists of multiple-stack frames. The state space $\mathcal{S}$ is either observable or unobservable (Silver et al., 2017; Zhang et al., 2020a). The agent uses observations $\mathcal{O}$ to sample actions from the action space $\mathcal{A}$. Every time the agent interacts with the environment, it obtains a reward $r$. The end goal is to train an agent to maximize the cumulative reward $\mathcal{R}$. The policy evaluation used as estimating the performance of the policy $\pi_\phi$ is normally defined by rewards in infinite-horizon tasks,

$$\mathbb{E}[\mathcal{R}] = \mathbb{E}\left[\sum_{t=0}^{\infty} \gamma^t r_t(\mathbf{s}_t, \mathbf{a}_t, \mathbf{s}_{t+1}) \,\Big|\, \pi_\phi\right]. \tag{1}$$

where $\gamma \in [0, 1)$ is the discount factor and $r_t$ denotes the reward at time $t$.

### 3.2 Q LEARNING

The state-action value function $Q_\theta$ is trained by minimizing the Bellman error to estimate the cumulative reward at the current state:

$$J_\theta(\mathcal{D}) = \mathbb{E}_{e \sim \mathcal{D}}[(Q_\theta(\mathbf{s}_t, \mathbf{a}_t) - r_t - \gamma Q_{\bar{\theta}}(\mathbf{s}_{t+1}, \pi_\phi(\mathbf{s}_{t+1})))^2] \tag{2}$$

where $e$ is a transition from the replay buffer $\mathcal{D}$. And $\bar{\theta}$ denotes an exponential moving average of $\theta$. For the continuous control tasks, we utilize an actor-critic algorithm called Deep Deterministic Policy Gradient (DDPG) (Silver et al., 2014; Lillicrap et al., 2015) which consists of the aforementioned state-action value function $Q_\theta$ and a deterministic policy $\pi_\phi$. The policy $\pi_\phi$ aims at maximizing $J_\phi(\mathcal{D}) = \mathbb{E}_{\mathcal{D}}[Q_\theta(\mathbf{s}_t, \pi_\phi(\mathbf{s}_t))]$. Various effective improvements have also been lead to DDPG. The Q-learning process incorporates $n$-step returns (Watkins, 1989; Peng & Williams, 1994). The scheduled exploration noise is produced by a linear decay $\tilde{\sigma}(t)$ for the variance $\tilde{\sigma}^2$ which provides different levels of exploration at different training steps: $\tilde{\sigma}(t) = \tilde{\sigma}_{\text{init}} + (1 - \min(t/T, 1))(\tilde{\sigma}_{\text{final}} - \tilde{\sigma}_{\text{init}})$. The initial and final value for standard deviation are defined by $\tilde{\sigma}_{\text{init}}$ and $\tilde{\sigma}_{\text{final}}$, and the decay horizon $T$ is related to the total training steps of the environment.

For the discrete control, we use the data-efficient Rainbow DQN (Van Hasselt et al., 2019) which applied multiple improvements on top of the original Nature DQN (Mnih et al., 2015).

### 3.3 STATE ABSTRACTION

While visual RL has achieved many successes in simulated tasks, it remains challenging to learn robust representations from real vision, where images reveal detailed scenes of a complex and unstructured world (Zhang et al., 2020b; Agarwal et al., 2021; Wang et al., 2022). Therefore, abstracting meaningful elements from the visual scene to present the underlying state is significantly important for visual RL.

We follow the Block Markov Decision Process (BMDP) (Du et al., 2019), which refers to episodic learning tasks via an unobservable latent space $\mathcal{S}$ and an observable context space $\mathcal{X}$. The environment generates a context by $\mathbf{x} \sim p(\cdot|\mathbf{s})$. They present a fundamental assumption as: each observation $\mathbf{x}$

uniquely determines its generating state $\mathbf{s}$. Similarly, in model-free RL without modeling dynamics, the manipulated context $\mathbf{x}$ can be conditioned on a certain probability given an environment transition $\mathbf{e}$ which is $\mathbf{x} \sim p(\cdot|\mathbf{e})$.

# 4 THE COIT

## 4.1 LEARNABLE INVARIANT TRANSFORMATION

Following the motivation of smoothing training experiences to stabilize the target $Q$ network (Mnih et al., 2013; Lillicrap et al., 2015), the transformed $\mathbf{x}'$ is required to satisfy $\mathbf{x}' \sim p(\cdot|\mathbf{e})$, where environment transition $\mathbf{e}$ is ideally in the replay distribution $\mathcal{D}$. Note that $\mathbf{e}$ is a random variables. Formally, we are ready to introduce the optimal invariant metric to reach the stationary distribution $\mathcal{D}$ over the augmented context $\mathbf{x}'$, through the definition,

**Definition 4.1.** (Optimal Invariant Metric). Given a transition distribution $\mathcal{D}$ for tuples in the replay buffer, suppose the block structure assumption holds, the shift between transitions $\mathbf{x}$ and its context $\mathbf{x}'$ can be measured by a conditional divergence:

$$d(\mathbf{x}, \mathbf{x}'|\mathbf{e}) \triangleq \mathbb{E}_{e \sim \mathcal{D}} \left[ d_{KL} \left( p(\mathbf{x}|\mathbf{e}=e) || p(\mathbf{x}'|\mathbf{e}=e) \right) \right] = \int d_{KL} \left( p(\mathbf{x}|\mathbf{e}=e) || p(\mathbf{x}'|\mathbf{e}=e) \right) dp(\mathbf{e}) \quad (3)$$

where $d_{KL}(\cdot||\cdot)$ is the Kullback-Leibler (KL) divergence. It indicates the expected distance between $\mathbf{x}$ and $\mathbf{x}'$ conditioning on $\mathbf{e}$. One may argue that the dynamics cannot be assumed as a fixed distribution when it comes to new observations, especially after data manipulation. Nevertheless, it generally claims that the experiences are uniformly sampled in a replay memory (Mnih et al., 2013). Also, the fact that the given conditions of the transition are consistent for the observed data as well as the transformed data, makes the above definition reasonable. Next, we will show why the conditional divergence defined in Eq.(3) is an optimal invariant metric from theoretical perspectives.

We employ the Bayes' rule on the conditionally distribution $p(\mathbf{x}|\mathbf{e}) = p(\mathbf{e}|\mathbf{x})p(\mathbf{x})/p(\mathbf{e}), \forall \mathbf{x} \in \mathcal{O}, \mathbf{e} \in \mathcal{D}$. Then the transition operator $p(\mathbf{e}|\mathbf{x})$ can be further divided as $p(\mathbf{e}|\mathbf{s})p(\mathbf{s}|\mathbf{x})$ for any $\mathbf{x} \in \mathcal{O}$, if $\mathbf{e}$ and $\mathbf{x}$ are conditional independent given $\mathbf{s}$[1]. Eq.(3) is rewritten as,

$$\mathbb{E}_{\mathbf{e}} \left[ d_{KL} \left( p(\mathbf{x}|\mathbf{e}) || p(\mathbf{x}'|\mathbf{e}) \right) \right] = \mathbb{E}_{\mathbf{e}|\mathbf{s}} \left[ d_{KL} \left( p(\mathbf{s}|\mathbf{x})p(\mathbf{x}) || p(\mathbf{s}|\mathbf{x}')p(\mathbf{x}') \right) \right] \quad (4)$$

Therefore, minimizing the conditional divergence leads to encoding the observation $\mathbf{x}$ and the transformed context $\mathbf{x}'$ into an invariant latent state space $\mathcal{S}$. As a consequence, the learnable pixel transformation is an optimality invariant combining a qualified encoder.

Now we have the observation encoder $g : \mathcal{O} \to \mathcal{S}$ mapping from the observed state $\mathcal{O}$ to the latent state $\mathcal{S}$ by a non-trivial function $g$ such that $g(\mathbf{x}) = p(\mathbf{s}|\mathbf{x}), \forall \mathbf{x}$. Traditionally, there is only one encoder dubbed feature backbone in RL models. Since the pixel transformation could drift away, e.g., supported by different components with those supporting $\mathbf{x}$ (Du et al., 2019). To enforce the invariant hidden states, another state encoder $g'$ that can map $\mathbf{x}'$ to $\mathbf{s}$ should exist, which is $g'(\mathbf{x}') = p(\mathbf{s}|\nu(\mathbf{x}, \cdot)), \forall \mathbf{x}', \nu(\mathbf{x}, \cdot)$ is the transformation.

So far, the goal of learning the optimal transformed data and encoders boils down to minimizing the distance between representations $g(\mathbf{x})$ and $g'(\mathbf{x}')$. To tackle the issue, we first provide two definitions to introduce measurements as follows,

**Definition 4.2.** ($\epsilon$-Approximation). Given a distance metric $d : \mathcal{O} \times \mathcal{S} \to \mathbb{R}_+$ satisfies $d(\mathbf{s}, \mathbf{s}) = 0, \forall \mathbf{s}$, and let $g, g' : \mathcal{O} \to \mathcal{S}$ be two functions. Let $\epsilon \geq 0$, given a distribution $\hat{\mathcal{D}}$ on $\mathcal{O}$, then $g$ and $g'$ are $\epsilon$-approximate w.r.t. $(d, \mathcal{D})$ if $\mathbb{E}_{\mathbf{x} \sim \hat{\mathcal{D}}} \left[ d(g(\mathbf{x}), g'(\mathbf{x})) \right] \leq \epsilon$.

**Definition 4.3.** ($\beta$-Similarity). Given a distance metric $d : \mathcal{O} \times \mathcal{S} \to \mathbb{R}_+$ satisfies $d(\mathbf{s}, \mathbf{s}) = 0, \forall \mathbf{s}$. There exists $g : \mathcal{O} \to \mathcal{S}$. Let $\beta \geq 0$, given distributions $\hat{\mathcal{D}}$ and $\hat{\mathcal{D}}'$ on $\mathcal{O}$, then $\mathbf{x}$ and $\mathbf{x}'$ are $\beta$-similar if $\mathbb{E}_{\mathbf{x} \sim \hat{\mathcal{D}}, \mathbf{x}' \sim \hat{\mathcal{D}}'} \left[ d(g(\mathbf{x}), g(\mathbf{x}')) \right] \leq \beta$.

Without loss of generality, the distance between the encoded states $g(\mathbf{x})$ and $g'(\mathbf{x}')$ can be expressed as the following triangular inequality. To obtain a metric, Kullback-Leibler divergence is rewritten in a form of the square root of Jensen-Shannon divergence. Therefore, we have,

$$d(g(\mathbf{x}), g'(\mathbf{x}')) \leq \underbrace{d(g(\mathbf{x}), g'(\mathbf{x}))}_{\text{encoding: } \epsilon\text{-Approximation}} + \underbrace{d(g'(\mathbf{x}), g'(\mathbf{x}'))}_{\text{augmentation: } \beta\text{-Similarity}} \quad (5)$$

---

[1] The tuples in reply buffer can be written as $\mathbf{e} = (\mathbf{s}_t, \mathbf{a}_t, r_t, \mathbf{s}_{t+1})$ after encoding, which makes $\mathbf{s}$ an intermediate random variable.

From the view of invariant learning, minimizing the right side of the inequality can upper bound our problem. The first term on the right side is the so-called $\epsilon$-approximation to measure the functional similarity after state abstraction, whereas the second term exists based on the procedure of data augmentation. Thus, we learn the encoders and shifted data simultaneously through the upper bound.

## 4.2 OPTIMAL STATE ABSTRACTION

To restrict the functional similarity of Eq.(5) from the perspective of learning a good encoding function with consistent semantics, the approaches formulating the main and momentum feature learning are utilized, motivated by contrastive learning (He et al., 2020). In particular, we enforce the encoding functions with exactly the same architecture, and use $\bar{\xi}^t = (1 - \tau_m)\bar{\xi}^{t-1} + \tau_m\xi^t$ at timestep $t$ to update the parameters of momentum function $g_{\bar{\xi}}$ with $g_{\xi}$, where $\tau_m \in [0, 1]$ is the updating rate. Furthermore, we design a projection that is $f : \mathcal{S} \to \mathcal{Y}$ using a ReLU network (Petersen & Voigtlaender, 2018) to upper bound the divergence by minimizing the distance in the projected space $\mathcal{Y}$. Previously, the projection has been proposed by Chen et al. (2020), while the theoretical guarantees of the underlying mechanism with momentum updating for model-free RL are explained in this work.

Suppose the Markov chain $\mathcal{O} \xrightarrow{g} \mathcal{S} \xrightarrow{f} \mathcal{Y}$ holds. For two functions $g$ and $f$ in the compatible ranges, we use $f \circ g$ to denote the function composition $f(g(\cdot))$. Before showing the proposed data transformed method, we introduce technical lemmas to take advantage of the designable projection function by leveraging the convexity. The momentum updating paradigm is capable of turning into momentum feature updating through a convex function or an equivalence of the convex function.

**Lemma 4.1.** Assume that $h : \mathbb{R}^{|\mathcal{S}|} \to \mathbb{R}^{|\mathcal{Y}|}$ can be written as $h(\xi) = f(<\xi, \mathbf{s}>)$, for some $\mathbf{s} \in \mathbb{R}^{|\mathcal{S}|}$, and $f : \mathbb{R}^{|\mathcal{Y}|} \to \mathbb{R}^{|\mathcal{Y}|}$ with parameter $\xi$. Then, convexity of $f$ implies the convexity of $h$.

**Lemma 4.2.** Given the dynamical updating: $\bar{\xi}^t = (1 - \tau_m)\bar{\xi}^{t-1} + \tau_m\xi^t$. By Lemma 4.1, $f_{\xi} = f_{\bar{\xi}}$ holds after convergence. As a result, the problem of $\min \mathbb{E}_{\mathbf{x}}[\|f_{\xi} \circ g_{\xi}(\mathbf{x}) - f_{\bar{\xi}} \circ g_{\bar{\xi}}(\mathbf{x})\|]$ is equivalent to the problem of $\min \mathbb{E}_{\mathbf{x}}[\|g_{\xi}(\mathbf{x}) - g_{\bar{\xi}}(\mathbf{x})\|]$.

To meet the requirement of a small upper bound, we state a theorem that provides some insights into why it is necessary to learn optimal transformed data together with the encoders.

**Theorem 4.1.** (CoIT) Suppose that Lipschitzness holds for functions $g_{\xi}, g_{\bar{\xi}}, f_{\xi}$ and $f_{\bar{\xi}}$, respectively. The updating dynamics is: $\bar{\xi}^t = (1 - \tau_m)\bar{\xi}^{t-1} + \tau_m\xi^t, \tau_m \in [0, 1]$. For any input $\mathbf{x} \sim \hat{\mathcal{D}}$ and transformed $\mathbf{x}'$ obtained via the transform operator $\nu(\mathbf{x}, \cdot)$, optimizing the conditional divergence in Definition 4.1 means to minimize the upper bound as follows,

$$\mathbb{E}_{\mathbf{x}}\left[d(f_{\bar{\xi}} \circ g_{\bar{\xi}}(\mathbf{x}), f_{\xi} \circ g_{\xi}(\mathbf{x}'))\right] \leq \rho \mathbb{E}_{\mathbf{x}}\left[\|\mathbf{x} - \mathbf{x}'\|\right] \tag{6}$$

where $\rho = L_g\left(CL_f + \|\xi_f\|\right), C = \frac{1+\tau}{1-\tau}, \tau = 1 - \tau_m$ are constants. $L_f$ and $L_g$ are Lipschitz constants of the functions $f(\mathbf{s})$ and $g(\mathbf{x})$, respectively.

The upper bound of the right side measures the margin of augmentation between original and transformed data. The left side measures the distribution changes in the latent space. Since both transformed data $\mathbf{x}'$ and encoder $g$ are updating, incorporating augmentation directly cannot well meet the basic stationary environment. Hereby, the theorem suggests us that the automatic transformation is used to bound the representation learning so that the abstracted states enhance the stationary distribution of tuples $e$ and facilitate efficient training. Proofs are given in Appendix A. The empirical comparisons of the projection network are presented in Appendix D.

## 4.3 PARAMETERIZABLE OBSERVATION

From the Theorem 4.1, we know the relation between the parameterized augmentation and the learnable latent state. That is, image transformation needs to be constrained by the distance between an observation and its associated augmentation. The optimal embedding can be obtained by minimizing of Eq.(5). Particularly, we parameterize the transformed data $\mathbf{x}'$ as $\nu(\mathbf{x}, \mathcal{G})$, where $\mathcal{G}$ is a Gaussian distribution dynamically changed along with the RL training.

In Algorithm 1, it defines the MDP $\mathcal{M}$ with Gaussian random variables $\mathcal{G}_0 \sim \mathcal{G}^{|\mathcal{O}|}$ for initialization. The TRANSFORM operator is fulfilled by the aforementioned pixel transformation $\nu$ which is a *shift*

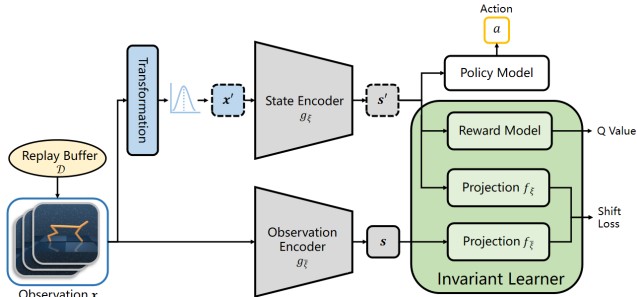

Figure 2: Overall architecture of CoIT. The observations are transformed following a Gaussian distribution $\mathcal{G}(\mu, \sigma)$ and encoded by the state encoder $g_\xi$. The observation encoder $g_{\bar{\xi}}$ and projection $f_{\bar{\xi}}$ are the exponentially moving average version of the state encoder $g_\xi$ and projection $f_\xi$.

subject following Gaussian distribution $\mathcal{G}_t(\mu_t, \sigma_t)$ on the top of data interpolation. The transformation $\mathbf{x}' = \nu(\mathbf{x}, \mathcal{G})$ is normalized according to both $\mathbf{x}$ and learned distribution $\mathcal{G}$, and thereby contributes to the cumulative reward maximization in an interactive way. We use the scope to the observation sampling for an optimal state abstraction. It can be regarded as data sampling from the replay buffer to reach a training-friendly distribution, dubbed *Contrastive Invariant Transformation*.

## 4.4 STABILIZING REWARD FUNCTION

It identifies one of the key impacts that CoIT in the RL training procedure as parametrizing the underlying invariant optimization to smooth the distribution $\mathcal{D}$ in the replay buffer. To further stabilize the reward function, we propose a mixed CoIT that samples multiple transformed data from the learned distribution $\mathcal{G}(\mu, \sigma)$, and then mix up the learned observation $\mathbf{x}'$. Similarly, we provide the invariant learning guarantee by optimizing the right side transformation in Theorem 4.2.

**Theorem 4.2.** (Mixed CoIT) Suppose that Lipschitzness holds for functions $g_\xi$, $g_{\bar{\xi}}$, $f_\xi$ and $f_{\bar{\xi}}$, respectively. The updating dynamics is: $\bar{\xi}^t = (1 - \tau_m)\bar{\xi}^{t-1} + \tau_m \xi^t$. For any input $\mathbf{x} \sim \hat{\mathcal{D}}$ and transformed $\mathbf{x}'$, the divergence with mixed transformed observation can be bound by,

---

**Algorithm 1** Parameterized Transformation in RL

---

1: **Initialization:** Draw distribution $\mathcal{G}_0 \sim \mathcal{G}^{|\mathcal{O}|}$ with given mean $\mu_0$ and variance $\sigma_0^2$; Set cumulative reward $\mathcal{R} = 0$.
2: **Training:**
3: **for** each timestep $t$ in $0, \cdots, T$ **do**
4:     $\mathbf{x}'_t = \text{TRANSFORM}(\mathbf{x}_t, \mathcal{G}_t)$
5:     $\mathcal{R} = \mathcal{R} + \gamma^t r(\mathbf{x}'_t, \mathbf{a}_t)$
6:     Adjust to an optimal $\mathcal{G}_t(\mu_t, \sigma_t)$.
7: **end for**

---

\* Details about learning $\mu_t$ and $\sigma_t$ are given in Algorithm 2 in Appendix B.

---

$$\mathbb{E}_{\mathbf{x}}\left[d(f_{\bar{\xi}} \circ g_{\bar{\xi}}(\mathbf{x}), f_\xi \circ g_\xi(\mathbb{E}_{\mathbf{x}'}[\mathbf{x}']))\right] \leq \rho \mathbb{E}_{\mathbf{x}} \mathbb{E}_{\mathbf{x}'}\left[||\mathbf{x} - \mathbf{x}'||\right] \tag{7}$$

where $\rho = L_g\left(CL_f + \|\xi_f\|\right), C = \frac{1+\tau}{1-\tau}, \tau = 1 - \tau_m$. $L_f$ and $L_g$ are Lipschitz constants of the functions $f(\mathbf{s})$ and $g(\mathbf{x})$ respectively. The proof of Theorem 4.2 is straightforward based on Theorem 4.1, and the details are presented in the supplement.

## 4.5 LEARNING CONTRASTIVE INVARIANT TRANSFORMATION

With theoretical analysis of invariant transformations, we presented a new framework with normalization variants to ensure above discussed learning guarantees by optimizing parameters. We initialize a distribution $\mathcal{G}_t(\mu, \sigma)$ and use the TRANSFORM operator to produce different views of $\mathbf{x}_t$ (Algorithm 1). The transformed data $\mathbf{x}'_t$ is viewed as the positive pair of $\mathbf{x}_t$. We also utilize a similarity metric $d$ to learn contrastive invariant transformation for the encoder $g_\xi(\cdot)$.

We first apply the bilinear interpolation to $\mathbf{x}_t$ and sample *shift* terms from $\mathcal{G}_t$ to produce multiple positive samples $\mathbf{x}_t^1, \mathbf{x}_t^2, .., \mathbf{x}_t^n$ and mix them together as $\mathbf{x}'_t$ following Theorem 4.2. To prevent the

distribution of transformed data from shifting too far away from the replay buffer $\mathcal{D}$, we borrow a similar idea from Yin et al. (2020) to regularize and smooth the distribution shift between the transformed and overall data. We use the statistical data stored in the BatchNorm layers to approximate the distribution of the overall data. In this way, the distribution shift between the transformed and overall data can be estimated by the following formulation,

$$\mathcal{K}_\omega(\mathbf{x}'_t) = \sum_l \left\| \tilde{\mu}(\mathbf{x}'_t) - \mathbb{E}(\tilde{\mu}_l(\mathbf{x})|\mathcal{O}) \right\|_2 + \sum_l \left\| \tilde{\sigma}^2(\mathbf{x}'_t) - \mathbb{E}(\tilde{\sigma}_l^2(\mathbf{x})|\mathcal{O}) \right\|_2 \tag{8}$$

where $\tilde{\mu}(\mathbf{x}'_t)$ and $\tilde{\sigma}^2(\mathbf{x}'_t)$ are the mean and variance of the transformed data and $\omega$ represents parameter collection $\{\mu_t, \sigma_t\}$ of the Gaussian distribution $\mathcal{G}_t(\mu, \sigma)$. The expectation terms $\mathbb{E}(\tilde{\mu}_l(\mathbf{x})|\mathcal{O})$ and $\mathbb{E}(\tilde{\sigma}_l^2(\mathbf{x})|\mathcal{O})$ respectively denote the estimation of the batch-wise mean and variance for the feature map corresponding to the $l$-th convolution layer, and $\mathcal{O}$ is the given observations.

Second, we utilize the similarity metric $d$ proposed by Chen et al. (2020) for learning the encoder $g_\xi(\cdot)$ which maps high-dimensional observation to embeddings to meet the invariant transformation in Eq.5. Given a positive observation pair $(\mathbf{x}_t, \mathbf{x}'_t)$, the loss is given by

$$\mathcal{L}_{\xi,\bar{\xi},\omega}(\mathcal{D}) \triangleq \left\| f_\xi(g_\xi(\mathbf{x}'_t)) - f_{\bar{\xi}}(g_{\bar{\xi}}(\mathbf{x}_t)) \right\|_2^2 = 2 - 2 \cdot \frac{\langle f_\xi(g_\xi(\mathbf{x}'_t)), f_{\bar{\xi}}(g_{\bar{\xi}}(\mathbf{x}_t)) \rangle}{\left\| f_\xi(g_\xi(\mathbf{x}'_t)) \right\|_2 \cdot \left\| f_{\bar{\xi}}(g_{\bar{\xi}}(\mathbf{x}_t)) \right\|_2} \tag{9}$$

Here $\bar{\xi}$ denotes the momentum version of parameters $\xi$ and $f_\xi(\cdot)$ is a non-linear projection of the representations embedded by $g_\xi(\cdot)$. $\mathcal{D}$ indicates the tuples stored in the replay buffer.

Next, we update the critic network $Q_\theta$ with transformed data $\mathbf{x}'_t$ and $\mathbf{x}'_{t+n}$ to minimize the TD error for $n$-steps returns. This is regarded as a regularized Q learning by Yarats et al. (2020) where the regularized representation learning is beneficial for optimal action taking (Zhang et al., 2020a).

$$J_Q(\mathcal{D}; \theta, \omega, \xi) = \left( Q_\theta\left(g_\xi(\mathbf{x}'_t), \mathbf{a}_t\right) - \sum_{i=0}^{n-1} \gamma^i r_{t+i} - \gamma^n Q_{\bar{\theta}}\left(g_\xi(\mathbf{x}'_{t+n}), \pi(\cdot|g_\xi(\mathbf{x}'_{t+n}))\right) \right)^2 \tag{10}$$

Eventually, we give the unified objective function as the full version of the CoIT,

$$J_{\theta,\xi,\omega}(\mathcal{D}) = J_Q(\mathcal{D}) + \alpha \mathcal{L}_{\xi,\bar{\xi},\omega}(\mathcal{D}) + \lambda \mathcal{K}_\omega(\mathcal{D}) \tag{11}$$

where $\alpha$ and $\lambda$ are hyper-parameters and the overall architecture is presented in Figure 2. We replace the vanilla Q-learning by $J_{\theta,\xi,\omega}(\mathcal{D})$ and the entire algorithm is presented in Algorithm 2 in Appendix B. Then, we evaluate CoIT on popular benchmarks to demonstrate the benefits of our method.

## 5 EXPERIMENTS

In this section, we benchmark our method on the DeepMind control suite and Atari100K. We compare CoIT with prior model-free methods first, then we present ablation studies to show the details of our method. Implementation details can be found in Appendix C.

### 5.1 ENVIRONMENTS

**DMControl.** DeepMind control suite (Tassa et al., 2018) is a widely used benchmark with several robot control tasks. Each episode is set to be $1,000$ frames and we use the total experienced frames to measure the data-efficiency. The per-frame reward is in the unit interval $[0, 1]$, so each episode contains a total reward of no more than $1,000$. Considering the different difficulties depending on tasks, we refer to setting more episodes with hard tasks for better evaluation.

**Atari100K.** There have been a number of prior papers that have benchmarked data-efficiency on the Atari 2600 Games for discrete control. Van Hasselt et al. (2019) and Kielak (2019) propose the data-efficient version of Rainbow DQN (Hessel et al., 2018) compared with human performance (Kaiser et al., 2019) within 100K time steps (400K frames, frame skip of 4). This sample-constrained evaluation is the so-called Atari100K and we benchmark CoIT on all 26 Atari Games.

### 5.2 BASELINES

**DMControl.** For continuous control we present several baselines, including methods of using data augmentation and contrastive learning to improve data-efficiency: (i) DrQ-v2 (Yarats et al., 2021), (ii)

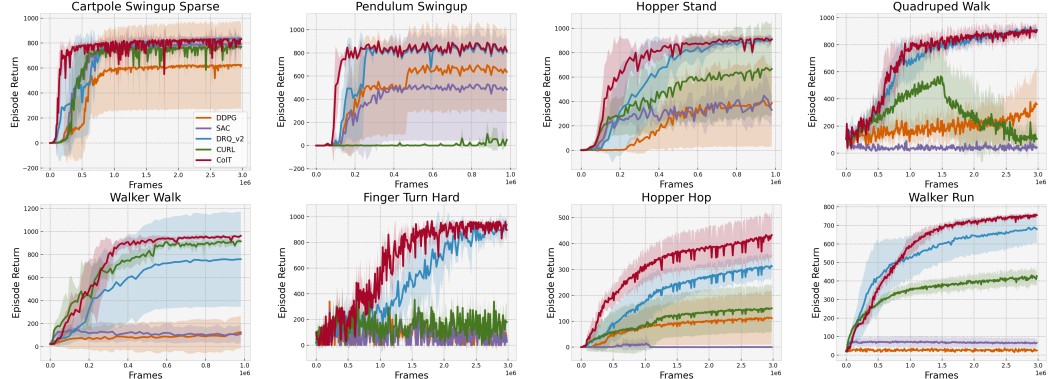

Figure 3: Results of complex tasks in DMControl. These tasks are chosen to offer multiple degrees of challenges, including complex dynamics, sparse rewards, hard exploration, and more.

CURL (Laskin et al., 2020b), (iii) Pixel SAC, and (iv) Pixel DDPG : Vanilla SAC and DDPG training directly from pixels. All methods are evaluated with the same periodicity of frames and average over 10 episodes return for evaluation query.

**Atari100K.** To benchmark the data-efficiency of CoIT for discrete control tasks, we compare our method to (i) DrQ (Yarats et al., 2020), (ii) CURL (Laskin et al., 2020b), (iii) SPR (Schwarzer et al., 2020), (iv) Random Agent, and Human Performance (Kaiser et al., 2019). All the algorithms are evaluated within 100K time steps for interaction. We average CoIT's performance over 10 random seeds and report the *best* score for each game following prior works.

## 5.3 MAIN RESULTS

**DMControl.** We choose 8 complex tasks from the DMControl for evaluation and present the results in Figure 3 and Table 2 in Appendix C. We also report the percentage (%) of score solved in the DMControl for baselines and CoIT in 500K and 100K steps in Figure 1. Below are key findings: (i) CoIT outperforms vanilla DDPG and SAC in a wide range. (ii) We also compare CoIT with DrQ-v2, a remarkable method for continuous control, to better demonstrate our method's data-efficiency. (iii) From general trends of the learning curves, CoIT improves or keeps the data-efficiency in a more stable manner which is not trivial on DMControl tasks.

**Atari100K.** We present results for Atari100K in Table 1 and below are key findings: (i) CoIT achieves top-performance on 10 of 26 games while still being competitive in the rest. (ii) CoIT *surpasses superhuman performance on 6 games* on the basis of Rainbow DQN . (iii) We also report the *mean* and *std* of the scores achieved by CoIT in 10 of 26 games which are top-performance. We present the results of two versions of CoIT (mixed & no-mixed) in Figure 6 in the appendix. From the histogram, we find that CoIT has much better stability, which is similar to the observation in the DMControl.

## 5.4 ABLATION STUDIES

We first visualize the TRANSFORM operator to demonstrate that there exists an invariant transformation for each task. We initialize the Gaussian distribution $\mathcal{G}_t(\mu, \sigma)$ based on the range of pixel shifts and plot the curves of the *mean* $\mu$ and *std* $\sigma$ during training in the DMControl in Figure 4.

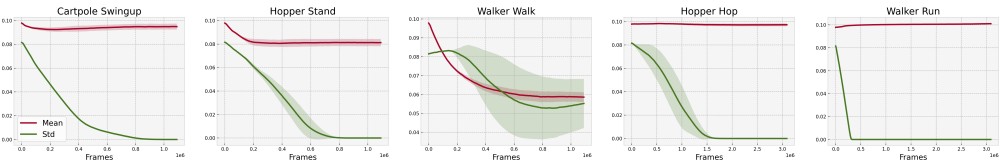

Figure 4: Visualization of the parameters of the Gaussian distribution for TRANSFORM.

Table 1: Mean episodic returns achieved by CoIT and baselines on 26 Atari games benchmarked at 100K environment steps. The results are recorded and averaged over 10 random seeds.

| Game | Human | Random | SimPLe | CURL | DrQ | SPR | CoIT |
|---|---|---|---|---|---|---|---|
| Aline | 7127.7 | 227.8 | 616.9 | 558.2 | 771.2 | 801.5 | **1206.7** |
| Amidar | 1719.5 | 5.8 | 88.0 | 142.1 | 102.8 | 176.3 | **182.3** |
| Assault | 742.0 | 222.4 | 527.2 | 600.6 | 452.4 | 571.0 | **635.7** |
| Asterix | 8503.3 | 210.0 | 1128.3 | 734.5 | 603.5 | **977.8** | 709.0 |
| Bank Heist | 753.1 | 14.2 | 34.2 | 131.6 | 168.9 | **380.9** | 124.8 |
| Battle Zone | 37187.5 | 2360.0 | 5184.4 | 14870.0 | 12954.0 | **16651.0** | 13760.0 |
| Boxing | 12.1 | 0.1 | 9.1 | 1.2 | 6.0 | **35.8** | 23.6 |
| Breakout | 30.5 | 1.7 | 16.4 | 4.9 | 16.1 | **17.1** | 16.1 |
| Chopper Command | 7387.8 | 811.0 | 1246.9 | 1058.5 | 780.3 | 974.8 | **1338.0** |
| Crazy Climber | 35829.4 | 10780.5 | **62583.6** | 12146.5 | 20516.5 | 42923.6 | 17538.0 |
| Demon Attack | 1971.0 | 152.1 | 208.1 | 817.6 | **1113.4** | 545.2 | 846.4 |
| Freeway | 29.6 | 0.0 | 20.3 | 26.7 | 9.8 | 24.4 | **29.6** |
| Frostbite | 4334.7 | 65.2 | 254.7 | 1181.3 | 331.1 | 1821.5 | **2069.8** |
| Gopher | 2412.5 | 257.6 | **771.0** | 669.3 | 636.3 | 715.2 | 746.8 |
| Hero | 30826.4 | 1027.0 | 2556.6 | 6279.3 | 3736.3 | 7019.2 | **7572.8** |
| Jamesbond | 302.8 | 29.0 | 125.3 | **471.0** | 236.0 | 349.0 | 336.0 |
| Kangaroo | 3035.0 | 52.0 | 323.1 | 872.5 | 940.6 | 3276.4 | **4116.6** |
| Krull | 2665.5 | 1598.0 | **4539.9** | 4229.6 | 4018.1 | 3688.9 | 3426.2 |
| Kung Fu Master | 22736.3 | 258.5 | **17257.2** | 14307.8 | 9111.0 | 13192.7 | 9250.0 |
| Ms Pacman | 6951.6 | 307.3 | 1480.0 | 1465.5 | 960.5 | 1313.2 | **1509.6** |
| Pong | 14.6 | −20.7 | **12.8** | −16.5 | −8.5 | −5.9 | 1.5 |
| Private Eye | 69571.3 | 24.9 | 58.3 | **218.4** | −13.6 | 124.0 | 145.7 |
| Qbert | 13455.0 | 163.9 | 1288.8 | 1042.4 | 854.4 | 669.1 | **2117.5** |
| Road Runner | 7845.0 | 11.5 | 5640.6 | 5661.0 | 8895.1 | **14220.5** | 11758.5 |
| Seaquest | 42054.7 | 68.4 | **683.3** | 384.5 | 301.2 | 583.1 | 554.0 |
| Up N Down | 11693.2 | 533.4 | 3350.3 | 2955.2 | 3180.8 | **28138.5** | 4734.2 |

According to the curves below, we find that the mean and std converge to an interval as the training goes on. These results demonstrate that the gaussian distribution proposed in CoIT could automatically find a `TRANSFORM` to smooth the distribution shift between the different views of the same observation, therefore being beneficial to the representation learning.

Then, we study the effects of different components in Eq.(11). This object function is composed of two parts: $\mathcal{K}_\omega(\mathbf{x}'_t)$ for regularization and $\mathcal{L}_{\xi,\bar{\xi},\omega}(\mathcal{D})$ for similarity metric. On this basis, we divide CoIT into 4 versions: (i) *Critic*. Transformation is only updated with the critic. (ii) *X-stats & Critic*. Transformation is updated by critic and $\mathcal{K}_\omega(\mathbf{x}'_t)$ together. (iii) *H-dist & Critic*. Transformation is updated by critic and $\mathcal{L}_{\xi,\bar{\xi},\omega}(\mathcal{D})$ together. (iv) *Unified Objective*. We evaluate all of these versions on 8 representative tasks from the DMControl and present the results in Figure 9 in the appendix.

Compared *Critic* with other variants, we demonstrate that both of the components are beneficial to the performance. Though *Critic* is data-efficient on most tasks, it may fall into trivial solutions. To solve this issue, we utilize the regularization in Eq.(8) with the similarity metric in Eq.(9) to meet the invariant transformation. Thus the *Unified Objective*'s performance leads ahead of all tasks. See Appendix C for extra ablation studies.

# 6    CONCLUSION

A novel pixel transformation CoIT under model-free RL algorithms that significantly improves the data-efficiency and stability for visual tasks is introduced in this work. We theoretically analyze how the learnable transformation constrains the distribution of transformed data, and dissect its benefits to representation learning. CoIT is no need for any additional modifications to the backbone RL algorithm and is easy to implement. We compare CoIT to SOTA methods on popular benchmarks and certify that it gains promising performance with advanced stability. Hopefully, contrastive invariant transformation can lead to a new branch for representation learning in RL.

## ACKNOWLEDGMENT

This work was supported in part by the National Key Research and Development Program of China No. 2020AAA0103400, National Key Research and Development Program of China No. 2021ZD0201504, National Natural Science Foundation of China under Grant 62273347, and CCF-Tencent Open Research Fund RAGR20220104. We thank anonymous reviewers for their discussions and feedback on the paper.

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
