# OpenReview forum: "On the Data-Efficiency with Contrastive Image Transformation in Reinforcement Learning"
_ICLR.cc/2023/Conference — ICLR 2023 poster_

### Official Review · Reviewer_RdJD · 2022-10-18

**Confidence:** 4
**Correctness:** 3
**Technical Novelty And Significance:** 2
**Empirical Novelty And Significance:** 2
**Recommendation:** 6

**Clarity, Quality, Novelty And Reproducibility:**

Clarity: The proposed idea is presented clearly.

Quality and Novelty: The idea is novel and the theoretical analysis is also provided. However, the contribution and improvement seem to be limited.

Reproducibility: The code is provided.

**Strength And Weaknesses:**

Strength:
1. The idea proposed in this paper is novel.
2. The proposed idea is presented clearly and the paper is globally well-written.
3. The theoretical analysis is provided.
4. The proposed framework is easy to implement.

Weakness:
1. Although the idea seems to be new, the contribution of this paper seems to be limited. According to Table 1, the improvement resulted from the proposed framwork is quite limited. Second, it is not discussed whether the proposed method can also contribute to improved performance in addition to sample efficiency. Besides, the improvement on stability is not very obvious, more explainations should be provided.
2. The paper lacks comparison with more recent methods, such as soda, svea and so on.
3. The paper lacks parameter analysis. For instance, how are hyperparameters α and λ determined?

Minor:
1. The details of how to adjust to the optimal Gt(µt, σt) can be mentioned in Algorithm 1.
2. Lf and Lg should also be mentioned in the paper.

**Summary Of The Paper:**

The paper proposes a new framework to improve the sample efficiency of model-free RL algorithms through combining them with a learnable data augmentation approach. The paper provides a theoretical analysis of how their method works to improve the representation learning and reports results on DMControl and Atari100K.

**Summary Of The Review:**

This paper proposes a new solution for the problem, and experiments on two benchmarks demonstrates its effectiveness. The major concerns are limited contributions and the lack of comparison with more recent approaches.

---

> ### Author Response · Authors · 2022-11-19
> **Repose to Reviewer RdJD [2/2]**
>
> ##### **Q3:** The improvement on stability is not very obvious, more explainations should be provided.
> **A:** There is no guarantee that random augmentation can meet the assumption about the stationary replay buffer. Though augmenting data extends the distribution of the replay buffer and improves the data-efficiency, it can also harm the training process due to diversifying the observations too much. Therefore, we consider designing a learnable transformation to stabilize the training.
>
> The results confirmed that our method is more stable on 4 tasks in the DMC (i.e., *Pendulum Swingup*, *Walker Walk*, *Finger Turn Hard*, *Walker Run* in Figure 3), and correspondingly provided evidence that CoIT leads to a much stationary replay buffer. For Atari games, we also plot the mean and variance of scores in 10 games where our method is better than all the baselines in Appendix C, Figure 7, to show the benefit of the mixed CoIT when facing complex and random dynamics.
>
> ##### **Q4:** The paper lacks comparison with more recent methods, such as soda, svea and so on.
> **A:** Thanks for your reminder, we have added the evaluation curves in the appendix compared to soda [2] and svea [3]. We also present the results at 500k steps below,
>
> | 500K Steps       | CoIT        | SODA        | SVEA        |
> | ---------------- | ----------- | ----------- | ----------- |
> | Cartpole Swingup | $866\pm14$  | $834\pm18$  | $824\pm31$  |
> | Hopper Stand     | $907\pm29$  | $674\pm163$ | $645\pm144$ |
> | Walker Walk      | $958\pm10$  | $890\pm22$  | $952\pm7$   |
> | Walker Run       | $585\pm47$  | $314\pm27$  | $433\pm38$  |
> | Hopper Hop       | $386\pm81$  | $153\pm109$ | $144\pm6$   |
> | Finger Turn Hard | $509\pm126$ | $101\pm75$  | $102\pm58$  |
>
> We implement these methods in DMControl and follow the framework and hyperparameters proposed in their papers. To be noticed, these methods use data augmentation with auxiliary tasks or *Q*-function regularized to learn invariant representations for policy transfer rather than data-efficiency improvement.
>
>
> ##### **Q5:** The paper lacks parameter analysis. For instance, how are hyperparameters α and λ determined?
> **A:** Due to the environment dynamics between tasks being similar in the DMC, keeping the hyperparameters fixed across tasks is enough for outstanding performance. However, we find that the dynamics and observation space among games in Atari are pretty different. Some games contain random dynamics to evaluate whether the agent memorizes a trajectory, while others focus on testing the ability of state abstraction. Therefore it is reasonable to search for optimal weights in each game.
>
> ##### **Q6:** Typos and unclear notations.
> **A:** We thank the reviewer for giving us thorough feedback regarding the presentation of the paper, we will incorporate the feedback in the next revision of the paper.
>
> [1] Kaiser, L., Babaeizadeh, M., Milos, P., Osinski, B., Campbell, R. H., Czechowski, K., ... & Michalewski, H. (2019). Model-based reinforcement learning for atari. arXiv preprint arXiv:1903.00374. [https://arxiv.org/abs/1903.00374]
>
> [2] Hansen, N., & Wang, X. (2021, May). Generalization in reinforcement learning by soft data augmentation. In 2021 IEEE International Conference on Robotics and Automation (ICRA) (pp. 13611-13617). IEEE. [https://arxiv.org/abs/2011.13389]
>
> [3] Nicklas Hansen, Hao Su, and Xiaolong Wang. Stabilizing deep q-learning with convnets and vision transformers under data augmentation. Advances in Neural Information Processing Systems, 34, 2021. [https://arxiv.org/abs/2107.00644]

---

> ### Author Response · Authors · 2022-11-19
> **Response to Reviewer RdJD [1/2]**
>
> Thank you for the constructive feedback and appreciation of our approach's novelty. We have revised the paper to include experiments with more recent methods in appendix. Here, we aim to address the questions raised in the review.
> ##### **Q1:** According to Table 1, the improvement resulted from the proposed framwork is  limited.
> **A:** It is hard to outperform in all games due to the diverse design of Atari games [1]. However, our method can hugely improve data efficiency on games that contain subtle dynamic changes in the observation space, such as *Alien*, *Assault*, *Freeway*, and so forth. The main issue may be the choice of the backbone RL algorithm, as we implement CoIT based on the Data Efficient Rainbow DQN, whereas SPR employs a modified DQN, which performs better in Atari with data augmentation. The combination of CoIT with SPR will be considered in our future work.
>
> ##### **Q2:** It is not discussed whether the proposed method can also contribute to improved performance in addition to sample efficiency.
> **A:** From Figure 3 we can see that our method eventually outperforms baselines on some tasks (i.e., *Walker Walk*, *Finger Turn Hard*, *Hopper Hop*, *Walker Run*). It indicates that CoIT obtains promising performance when facing challenging tasks under limited training steps.
> Our main idea is to design an invariant transformation learning objective that benefits to data-efficiency. Although numerous methods can converge to an optimal policy under infinite training steps, this will lead to catastrophic training costs, which is not conducive to implementing the RL algorithm in a complex scene like the real world.

---

### Official Review · Reviewer_ao8s · 2022-10-24

**Confidence:** 3
**Correctness:** 3
**Technical Novelty And Significance:** 2
**Empirical Novelty And Significance:** 3
**Recommendation:** 6

**Clarity, Quality, Novelty And Reproducibility:**

The novelty and quality of the paper are relatively good. But the clarity needs to be improved, as summarized in "Strength And Weaknesses".

For reproducibility, a Github repo is provided, although I have not tried to run it yet.

**Strength And Weaknesses:**

### Strengths

- The idea of learnable augmentation is relatively novel.
- The authors provide some theoretical interpretation for why learnable augmentation is desired.
- The proposed method achieves relatively promising results on several benchmark environments.

### Weaknesses

- I think the proposed algorithm seems simple yet effective. I like the idea of learning augmentations. However, the theoretical analysis sections are a little confusing and not very to the point. First, why is the replay buffer fixed? During training of model-free RL, I would expect the replay buffer to be consistantly updated, with imperfect data coverage. It is weird to me that the paper motivates by "dynamic distribution" and "training-friendly" but does not discuss much on the distribution shift problem during training. Second, I don't think Section 4.1 and 4.2 have provided sufficient evidence for the "optimality" of the proposed metric and the algorithm. Some claims are not very convincing. For example, why does Theorem 4.1 suggest that "incorporating augmentation directly cannot well meet the basic stationary environment"?
- There exist a bunch of grammar errors and typos, which may cause confusion. Some notations are not clearly defined in the context. For example, in Theorem 4.1, what does "Lemma 4.2 holds for functions $f_{\xi}$ and $f_{\bar{\xi}}$ respectively" mean? Lemma 4.2 is a lemma, not a condition for $f$. Also, $L_g$ and $L_f$ appear in the theorem without definitions.
- The ablation study also confuses me. Why does the convergence of $\mu$ and $\sigma$ mean that TRANSFORM successfully smooths the distribution shifts? Can the authors provide more explanation?

Questions:
- Why is Gaussian distribution used to generate augmentations? Would a deconvolutional network work here?
- How is "tasks solved" defined in Figure 1?

**Summary Of The Paper:**

This paper presents a learnable data augmentation framework for visual RL, which learn the parameterized image augmentation together with model-free RL. Some theoretical interpretations are provided to show the rationale of learned augmentation. Experiments on DMC and Atari100 benchmarks demonstrate that the proposed method CoIT achieves better performance than SOTA augmentation methods for model-free RL.

**Summary Of The Review:**

This paper presents a relatively novel idea and show good empirical success of the method. But there are several clarity issues of the theory and critical claims, making it hard to justify the quality of the paper. I would like to increase my rating if the authors can address my confusion with convincing evidence.

---

> ### Author Response · Authors · 2022-11-19
> **Response to Reviewer ao8s**
>
> Thank you for your review. In the following, we will address your comments and hope that this clarifies the context of our work.
> ##### **Q1:** The reason for fixed replay buffer.
> **A:** Inevitably, the training data may not meet the i.i.d assumption when directly incorporating random augmentation (e.g., an excessive margin of augmentation like *random convolution* discussed in SVEA [1] will cause an unstationary replay buffer which is harmful to Q learning and representation learning). Also, the replay buffer used in DrQ-v2 is 10 times larger than that used in DrQ, which means that a larger and more stationary replay buffer leads to better data coverage and is beneficial for catching the distribution of the replay buffer. With the above states, we did not provide further evidence about the "dynamic distribution" when discussing the fixed replay buffer.
>
> ##### **Q2:** The evidence for the "optimality" of the proposed metric and the algorithm is insufficient.
> **A:** Although the *optimality* of representation learning in RL has been discussed in DrQ, where they use the Q function to illustrate the *Optimal Invariant Metric*, we provide evidence besides the Q function from a theoretical perspective and use Sections 4.1 & 4.2 to guide the architecture and the distribution of the transformation. Directly incorporating augmentation extends the replay buffer and can regularize the network, but an over-unstable replay buffer harms the training process. It is hard to say which augmentation is suitable for data efficiency and how to adjust the margin of the augmentation. Therefore we turn it into a learnable operator and try to provide a finer learning objective. Also, a learnable transformation is easier to meet the basic assumption about the stationary environment comparing naively incorporating augmentation.
>
> On the other hand, updating the encoder with transformation parallel requires a series of theoretical conditions to meet Equation 6. When Theorem 4.1 does not exist, there will be a situation like $\mathbb{E} [d(f_{\bar{\xi}} \circ g_{\bar{\xi}}(\mathbf{x}),f_{\xi}\circ g_{\xi}(\mathbf{x}^{\prime}))]> \rho\mathbb{E}_{\mathbf{x}}[\left \| \mathbf{x}-\mathbf{x}^{\prime} \right \| ]$, which may lead to large divergence.
>
> ##### **Q3:** What does "Lemma 4.2 holds for functions $f$ and $f$ respectively" mean? Lemma 4.2 is a lemma, not a condition for $f$.
> **A:** We have revised the descriptions in theorems. Considering the requirements in architecture design, the property of $f$ needs to be disscussed, which is prepared for the theoretical analysis of learning an optimal transformation.
>
> ##### **Q4:** More explanations about the convergence of transformation for smoothing the distribution shifts.
> **A:** It is clear to find in the results in Figure 4 that the distribution is dynamically converging. It indicates that our method samples data from a specific distribution, which consists of our idea in Equations 6 and 7, where the distance between x and x' in latent and observation space is minimized together. It also indicates that a fixed transformation or a random augmentation is insufficient compared to a learnable transformation that finds a transformed distribution automatically.
>
> ##### **Q5:** Why is Gaussian distribution used to generate augmentations?
> **A:** Theoretically, any parameterized data transformation can be used to meet our method. For example, a convolutional network with $1\times 1$ reparameterized kernels or a reparameterized deconvolutional network may generate augmentations. However, due to the complexity of the learning algorithm, we choose Gaussian distribution for a more friendly training.
>
> ##### **Q6:** How is "tasks solved" defined in Figure 1?
> **A:** According to the reward system designed in the DMC, the per-frame reward is in the unit interval [0,1]. Thus each episode contains a reward of no more than 1,000. When the agent's performance approaches the upper bound, we consider that the agent can solve the task. It is also described in other papers [2,3].
>
> [1] Nicklas Hansen, Hao Su, and Xiaolong Wang. Stabilizing deep q-learning with convnets and vision transformers under data augmentation. Advances in Neural Information Processing Systems, 34, 2021. [https://arxiv.org/abs/2107.00644]
>
> [2] Denis Yarats, Rob Fergus, Alessandro Lazaric, and Lerrel Pinto. Mastering visual continuous control: Improved data-augmented reinforcement learning. arXiv preprint arXiv:2107.09645, 2021. [https://arxiv.org/abs/2107.09645]
>
> [3] Hafner, D., Lillicrap, T., Ba, J., & Norouzi, M. (2019). Dream to control: Learning behaviors by latent imagination. arXiv preprint arXiv:1912.01603. [https://arxiv.org/abs/1912.01603]

---

> > ### Comment · Reviewer_ao8s · 2022-12-07
> > **Thank you for the response**
> >
> > My concerns have been mostly addressed, so I have increased my rating from 5 to 6.

---

### Official Review · Reviewer_yPYT · 2022-11-03

**Confidence:** 3
**Correctness:** 3
**Technical Novelty And Significance:** 2
**Empirical Novelty And Significance:** 2
**Recommendation:** 6

**Clarity, Quality, Novelty And Reproducibility:**

The clarity of the main theorem is not clear to me, and I expect more explanations from authors . Other parts are good.

There are several typos and wrong links in the paper, which might hurt the paper quality.

The novelty is limited but I appreciate the efforts that make it work.

The reproducibility is good since the authors provide the source code.

**Strength And Weaknesses:**

Strength:

- The idea is simple but effective, which might be easy to implement and follow.

- The authors make theoretical efforts in justifying the framework.
- The experiments are extensive, conducted on DMControl and Atari 100k. Also the baselines (DrQ v2 and CURL) are decent.

Weakness:

- The theoretical analysis in this paper uses several inequalities to achieve the final objective, while some reasoning looks not very clear to me. See following comments for more details.
- In Equation 5,  the authors use the triangular inequality $d(g(x),g'(x'))\leq d(g(x),g'(x))+d(g'(x),g'(x'))$ so that they could minimize the upper bound. However, the LHS $d(g(x),g'(x'))$ could be optimized directly. So what is the necessarity of using the upper bound? And it is better that the necessarity be illustrated  from both the theoretical perspective and the empirical perspective. I am very willing to take feedbacks from authors.
- In Equation 6, the authors still try to prove that minimizing the LHS means minimizing the RHS. However, the LHS is the lower bound and thus it seems not meaningful to minimize the lower bound. If I do not misunderstand, the main theorem might not be able to motivate the CoIT algorithm. There is a chance I misundertand and there could be more efforts in detailing the theorem.
- The writing could be improved. There are many variables not well stated. For example,  in Theorem 4.1, the variables L_g and L_f are not stated in the main paper.

- The authors propose a multi-task objective that combine three loss functions in total. Though it is common that different objectives are used with different weights, the hyperparameters given in appendix seem to show that the weight for objectives must be carefully choosen for each tasks to make it work. More explanations about the hyperparameter selection and the experiment observation about the hyperparameter might be necessary.



Typos:

- related work: DRQ -> DrQ
- main results: Table 2 -> Table 1
- Some theorem links are not correct. One careful check might be great.

note: this is an emergency review

**Summary Of The Paper:**

This paper proposes a simple yet effective learning module upon the framework of contrastive reinforcement learning. The module could automatically learn a optimal transformation for data augmentation. The algorithm is theoretically founded and also demonstrated by expeiments on DMControl and Atari 100k.

**Summary Of The Review:**

I like the proposed algorithm for its simplicity and the theoretical analysis is also somewhat interesting. The main concern for me is (a) whether the main theorem is constructive for the algorithm and (b) whether the objectives really help generally for different tasks or it is carefully tuned for each single task to make it work. The detailed questions are mentioned before, which I hope the authors could address.

---

> ### Author Response · Authors · 2022-11-19
> **Response to Reviewer yPYT**
>
> Thank you for your thoughtful review and constructive feedback. We have revised the paper with a new discussion based on your feedback, and we respond to individual points below.
> ##### **Q1:** What is the necessity of using the triangular inequality in Equation 5.
> **A:** Equation 5 is required for the proof of Theorem 4.1. Similar to PSM [1] where it provides a bound on policy transfer between the state $y\in \mathcal{Y}$ and its closest match in $\mathcal{X}$ by $\tilde{x}_y:=\text{argmin}_\mathcal{X} d^*(x,y)$:
>
> $\mathbb{E}  [\sum_{t\ge 0}{\gamma^t TV(\tilde{\pi}(Y_y^t),\pi (Y_y^t))}  ]\le \phi\cdot d^*(\tilde{x}_y,y)$, $\phi=\frac{1+\gamma}{1-\gamma}$
>
> Normally, the closest match $\tilde{x}_y$ is obtained by augmentation. In the sense of learning meaningful state abstraction, we use a similar way to illustrate the objectives can optimize both sides of the inequality with image transformation and encoding function simultaneously, though PSM focuses on behavior similarity to measure the representations in latent space.
>
> Our goal is to optimize $\mathbf{x}^{\prime}$ and $g^{\prime}$. The main idea to present the triangular inequality, in other words, is to indicate that optimizing the transformation with the encoder together is equal to minimizing the distance between encoders $g(\mathbf{x})$ with $g^\prime(\mathbf{x})$ and the distance between $\mathbf{x}$ with $\mathbf{x}^\prime$. To be noticed, the boundary of the solution space is wide when $\mathbf{x}^{\prime}$ and $g^{\prime}$ are updating together. Therefore we need mathematical formulas to illustrate the relations of image transformation and encoding function, and use them as a sort of constraint for the model-free RL algorithm. In Equation 6 and Equation 7, we show that the left side of the inequality in Equation 5 is directly optimized, and the representations in the latent space with the distance between $\mathbf{x}$ and $\mathbf{x}^{\prime}$ are minimized together during training.
>
> From an empirical perspective, we compare contrastive image transformation in the ablation study (see Figure 5). It shows that different transformation does matter for data-efficiency. The triangular inequality provides guidance about how to learn a qualified transformation so as to adjust the augmentation. By comparing with CURL, we would like to point out that Equation 5 is useful in decoupling encoding and augmentation, and subtle design for each procedure works eventually.
>
> ##### **Q2:** The LHS is the lower bound and thus it seems not meaningful to minimize the lower bound in Equation 6, thus it may not be able to motivate the CoIT algorithm (might be misunderstand).
> **A:** Here we use the inequality to discuss the conditions of the representation learning rather than an objective for optimization. Indeed, minimizing the left side does not equal minimizing the right side in Equation 6, as we aim to optimize both sides together. Our goal is to guarantee the diversity of $\mathbf{x}^{\prime}$ with the invariance of transformation to achieve consistent semantics, as Section 1 presents. (i.e., the margin of the diversity of $\mathbf{x}^{\prime}$ to meet this invariant transformation).
>
> ##### **Q3:** Details about the hyperparameter selection.
> **A:** The model selection in Atari proceeds due to the significant differences among games in Atari. In particular, games emphasize various aspects of policy evaluation, so it is reasonable to search for the optimal hyperparameter game-by-game. Moreover, methods with auxiliary components are often good at different kinds of games. For instance, SPR [2] gains outstanding scores when facing games with sequential dynamics, like *crazy climber* with the proposed multi-step prediction. Our method performs well in detecting changes in a semantic situation from observations. Thanks to DMC's less randomness and less complex observation space, we utilize a consistent set of hyperparameters and earn data-efficiency across multiple tasks.
>
> ##### **Q4:** Typos and unclear notations.
> **A:** Thanks for the suggestions. We have already corrected the aforementioned mistakes.
>
> [1] Rishabh Agarwal, Marlos C Machado, Pablo Samuel Castro, and Marc G Bellemare. Contrastive behavioral similarity embeddings for generalization in reinforcement learning. arXiv preprint arXiv:2101.05265, 2021. [https://arxiv.org/abs/2101.05265]
>
> [2] Max Schwarzer, Ankesh Anand, Rishab Goel, R Devon Hjelm, Aaron Courville, and Philip Bachman. Data-efficient reinforcement learning with self-predictive representations. arXiv preprint arXiv:2007.05929, 2020 [https://arxiv.org/abs/2007.05929]

---

> > ### Comment · Reviewer_yPYT · 2022-12-07
> > **Response to Authors**
> >
> > Thank authors for the feedback on addressing my issues. After carefully reading replies from authors, I may have the following questions remained.
> >
> > **More about Q1:** As mentioned by authors, "To be noticed, the boundary of the solution space is wide when $x'$ and $g'$  are updating together. Therefore we need mathematical formulas to illustrate the relations of image transformation and encoding function, and use them as a sort of constraint for the model-free RL algorithm."
> >
> > Could authors explain more why the boundary of the solution space is wide when $x'$ and $g'$  are updating together and why it could be narrowed when updating them separately? And from my personal perspective, the empirical implementation for the left side of the equation could not be very difficult, and  thus it might be good to see some empirical results.
> >
> > **More about Q3**: I certainly agree with the authors about the necessity of grid search for optimal hyperparameters. So could authors provide some more detailed experiment results for these searched parameters? It might be good for readers and reviewers to gain a sense of the effect of learned transformation.

---

### Official Review · Reviewer_7fsn · 2022-11-04

**Confidence:** 4
**Correctness:** 4
**Technical Novelty And Significance:** 3
**Empirical Novelty And Significance:** 4
**Recommendation:** 8

**Clarity, Quality, Novelty And Reproducibility:**

The paper is well written and easy to follow. The supplementary material has ample implementation details for reproducible. While the core idea of data augmentation for visual RL is not new, the proposed method takes that further and produces state of the art results.

**Strength And Weaknesses:**

Strengths:

1. While practical the proposed method is also justified from a theoretical perspective as well.
2. The approach is intuitive and straightforward.
3. Produces state-of-the art results on some standard benchmarks.

Weaknesses:

Only wandering what results are on the Humanoid task, which has now been successfully solved from images.

**Summary Of The Paper:**

The paper proposes a learnable data augmentation procedure for speeding up reinforcement learning from pixels. In particular in applies a transformation to the input images, which is parameterized by a Gaussain distribution. The proposed method combines three main components:

1. Standard Q-learning approach with data augmentation, similar to DrQ
2. Contrastive learning loss for the encoder
3. Smoothing objective for the transformation parameters

The paper proposes a theoretical justification for this approach and shows state-of-the-art results on DM Control and Atari.

**Summary Of The Review:**

Well written, intuitive paper with good justification and results.

---

> ### Author Response · Authors · 2022-11-19
> **Response to Reviewer 7fsn**
>
> Thank you for the constructive feedback, suggestions, and pointers. We also appreciate the acknowledgment of our approach's novelty. Here, we aim to address the questions raised in the review.
>
> ##### **Q1：** Evaluating agents by images on demanding tasks such as Humanoid.
> **A:** Thanks for your constructive comments and support. Although our method improved sample-efficient in several tasks, both for continuous and discrete control tasks, there still exists room for improvement in the Humanoid task. Tasks such as Humanoid are highly complex and cost a significant time for one run. Therefore, we need more time to tune the parameters. Additionally, it is necessary to modify the backbone model-free RL algorithms for better policy optimization to solve these complex continuous control tasks more efficiently, such as efficient state exploration and task-specific pose prediction for robots. We would like to do an experimental analysis of the direction in future works.

---

### Decision · Program_Chairs · 2023-01-20

**Decision:**

Accept: poster

**Justification For Why Not Higher Score:**

There are still some unresolved questions from the reviewers.

**Justification For Why Not Lower Score:**

N/A

**Metareview: Summary, Strengths And Weaknesses:**

The paper presents a new data augmentation approach, which is learnable, to improve visual reinforcement learning. Reviewers generally find the approach intuitive, effective and novel. There are some concerns on paper presentation and comparisons, but mostly got resolved during rebuttal. The AC agrees with the majority of the reviews and recommend to accept the paper.

**Note From Pc:**

if the above contains the word "oral" or "spotlight" please see: "oral" presentation means -> notable-top-5% and "spotlight" means -> notable-top-25%. As stated in our emails, we are disassociating presentation type from AC recommendations